# Sub-lethal effects of the consumption of *Eupatorium buniifolium* essential oil in honeybees

Carmen Rossini[1]*, Federico Rodrigo[1], Belén Davyt[1,2], María Laura Umpiérrez[1,2], Andrés González[1], Paula Melisa Garrido[2], Antonella Cuniolo[2], Leonardo P. Porrini[2], Martín Javier Eguaras[2], Martín P. Porrini[1,2]

**1** Laboratorio de Ecología Química, Facultad de Química, Universidad de la República de Uruguay, Montevideo, Uruguay, **2** Centro de Investigación en Abejas Sociales (CIAS), Instituto de Investigaciones en Producción Sanidad y Ambiente (IIPROSAM), Facultad de Ciencias Exactas y Naturales, Universidad Nacional de Mar del Plata, Mar del Plata, Buenos Aires, Argentina

* crossini@fq.edu.uy

**Data Availability Statement:** All relevant data are within the paper and its Supporting information files.

## Abstract

When developing new products to be used in honeybee colonies, further than acute toxicity, it is imperative to perform an assessment of risks, including various sublethal effects. The long-term sublethal effects of xenobiotics on honeybees, more specifically of acaricides used in honeybee hives, have been scarcely studied, particularly so in the case of essential oils and their components. In this work, chronic effects of the ingestion of *Eupatorium buniifolium* (Asteraceae) essential oil were studied on nurse honeybees using laboratory assays. Survival, food consumption, and the effect on the composition of cuticular hydrocarbons (CHC) were assessed. CHC were chosen due to their key role as pheromones involved in honeybee social recognition. While food consumption and survival were not affected by the consumption of the essential oil, CHC amounts and profiles showed dose-dependent changes. All groups of CHC (linear and branched alkanes, alkenes and alkadienes) were altered when honeybees were fed with the highest essential oil dose tested (6000 ppm). The compounds that significantly varied include n-docosane, n-tricosane, n-tetracosane, n-triacontane, n-tritriacontane, 9-tricosene, 7-pentacosene, 9-pentacosene, 9-heptacosene, tritriacontene, pentacosadiene, hentriacontadiene, tritriacontadiene and all methyl alkanes. All of them but pentacosadiene were up-regulated. On the other hand, CHC profiles were similar in healthy and *Nosema*-infected honeybees when diets included the essential oil at 300 and 3000 ppm. Our results show that the ingestion of an essential oil can impact CHC and that the effect is dose-dependent. Changes in CHC could affect the signaling process mediated by these pheromonal compounds. To our knowledge this is the first report of changes in honeybee cuticular hydrocarbons as a result of essential oil ingestion.

## Introduction

Homeostasis in honeybee colonies depends, among other factors, on semiochemicals that serve as key social regulators in *Apis mellifera*. These include pheromones such as cuticular

**Funding:** The authors would like to acknowledge financial support from the Comisión Sectorial de Investigación Científica, Universidad de la República-Uruguay (Programa Grupos, https://www.csic.edu.uy/, Grants 2018_2004 and 2014_980, Grupo 1734), Programa de Desarrollo de las Ciencias Básicas (PEDECIBA, Uruguay, http://www.pedeciba.edu.uy/indice.php; financial aid to CR), Agencia Nacional de Investigación e Innovación de Uruguay (Grant INNOVAGRO FSA_1_2013_12956 and Grant MRC_C_2011_1_7 from Programa Regional MERCOSUR educativo, Argentina/Uruguay; https://www.anii.org.uy/inicio/ ), and Agencia Nacional de Promoción Científica y Tecnológica (ANPCyT, Argentina; Fondo para la Investigación Científica y Tecnológica (FONCyT, Grant PICT-2014-0394; https://www.argentina.gob.ar/ciencia/agencia/fondo-para-la-investigacion-cientifica-y-tecnologica-foncyt). The funders had no role in study design, data collection and analysis, decision to publish, or preparation of the manuscript.

**Competing interests:** The authors have declared that no competing interests exist.

hydrocarbons (CHC), known to mediate social recognition in honeybees [1]. Semiochemicals in *A. mellifera* may be affected by many factors including environmental cues [2], the presence of the gut parasite, *Nosema ceranae* [3], and the use of in-hive acaricides to control *Varroa destructor* (Mesostigmata: Varroidae) (unpublished results). *Varroa destructor* is an ectoparasite that may contribute to the honeybee colony collapse disorder [4]. As a result of a prospecting program [5, 6] for botanical acaricides [7] based on essential oils (EO), we selected the EO from aerial parts of *Eupatorium buniifolium* (Asteraceae) as a candidate for follow-up studies. The vapors from the EO of *E. buniifolium* (hereafter EOEb) showed fumigant activity against varroa mites with no topical and fumigant toxicity for honeybees in laboratory assays, as well as a moderate acaricidal activity in preliminary field trials [5, 6]. These results prompted us to further investigate the potential sublethal effects of this EO on honeybees when supplied in the diet.

It is well documented that several agrochemicals [8–10] and even in-hive acaricides [11–13] cause sublethal effects on honeybees and other pollinators. Among these effects, subtle but ecologically relevant changes have been reported on the cognitive functions of bees [12, 14–23]. Neonicotinoid pesticides (imidacloprid, acetamiprid, thiamethoxam, etc.) are among the most well studied in this regard [8, 10, 21]. Exposure to imidacloprid by itself [19] and in combination with the acaricide coumaphos [16], as well as exposure to thiamethoxam [23], cause behavioral changes and impair the learning and olfactory memory processes of bees. Exposed forager bees show a reduced response in the proboscis extension reflex assay [19], indicating difficulties in the learning of aroma cues that predict a floral reward. Exposure to and oral consumption of acetamiprid [16, 24] also cause a negative effect on the associative memory of bees. Other sub-lethal effects on honeybee performance indicators such as reproduction [17] and larval development [25] have also been described when honeybees are exposed to pesticides. In the case of the herbicide glyphosate, topical contact impairs the ability of honeybees to return to their colonies [15], and affects the transcriptional levels of several genes [26]. Synthetic and natural acaricides, including thymol and formic acid usually used in organic practices, also affect the memory of honeybees [12]. Therefore, the need to consider these and other sublethal effects when registering new pesticides has been strongly advocated [14]. Furthermore, the study of chronic exposure effects is necessary if a plant product will be used to develop new botanical pesticides [7], as it is for conventional pesticides.

Products containing compounds characteristic of EO are currently being used as varroacides, but few studies have addressed the long-term effects of this type of products on honeybee performance [27]. In this study, the chronic effects resulting from the oral administration of EOEb were studied on honeybee nurses under laboratory conditions [28], assessing survival, food consumption and effects on the CHC profiles.

## Materials and methods

### Plant material and essential oil production

The aerial vegetative parts of *E. buniifolium* were collected in Las Brujas, Canelones (34.38° S, 56.20° W); in an experimental crop field of INIA (Instituto Nacional de Investigación Agropecuaria, Uruguay), during the summers of 2009–2010 and 2014–2015. Plant materials were identified by Prof. Eduardo Alonso-Paz (Cátedra de Botánica) and voucher specimens were deposited at the Herbarium of Facultad de Química, Montevideo, Uruguay (*E. buniifolium*: Santos s/n MVFQ 4391). The EOs were obtained by hydro-distillation by exogenously-generated steam distillation using a 200-L alembic connected to a 50-L plant material container (EYSSERIC). In all cases, after drying with anhydrous magnesium sulfate, EOs were stored in amber glass containers under nitrogen at -4°C. The EOs used in this study will be referred to

s EOEb_2009 (EO from *E. buniifolium* harvested in the summer of 2009–2010) and EOEb_2014 (EO from *E. buniifolium* harvested in the summer of 2014–2015).

## Bioassay with honeybees

**Honeybee husbandry and collection.** Experiments were performed using hybrid honeybees of *Apis mellifera mellifera* and *Apis mellifera ligustica* from colonies located at the experimental apiary of the Social Honeybees Research Centre (38˚10'06"S, 57˚38'10"W), Mar del Plata, Argentina. Colonies used to obtain brood combs were selected based on their low abundance of *N. ceranae* ($< 5 \times 10^5$ spores/bee [29]) and their low prevalence of *V. destructor* mites. The mite prevalence was evaluated by the natural mite fall method [30] and the rates of phoretic mite infestation [31]. Furthermore, neither of the colonies used to obtain the imagoes presented any visible clinical symptoms of other diseases (*i.e.*, American Foulbrood, Chalkbrood or viruses).

To avoid the presence of long lasting acaricide residues that are commonly found in new commercial beeswax [32], plastic foundations covered with a thin layer of virgin organic wax were used. These foundations were placed in the selected colonies three months before the onset of the assays and served as brood combs. For every assay, frames of sealed brood were transported in thermic boxes and maintained under controlled conditions until imago emergence (34˚C; 60% rH). Newly emerged honeybees were placed in groups of 200–300 in wooden cages (11 × 9 × 6 cm) with a plastic mesh in an incubator (28˚C; 40% rH). To constitute the replicates in each experiment, groups of 2-day-old honeybees (N = 30 to 50 imagoes/ group) were caged in transparent and ventilated plastic cylindical flasks (900 cm³) with a removable side door flasks and with inputs for gravity feeding devices [28]. These flasks were maintained in an incubator (28˚C; 40% rH) through the assays.

Honeybees were fed *ad libitum* with syrup (sucrose-water solution; 2:1 w:v) and fresh beebread (stored pollen) collected from combs from the original hive. Besides, in the case of the experimental flasks, water and one of the different treatment diets (Eos prepared in ethanol, see below section 2.3) diluted in the sucrose syrup were also provided through the assays. Except for pollen (replaced weekly), all diet items were replaced daily.

**Diet preparation.** To prepare the diets, Eos were dissolved in EtOH 96% and then the solutions were added to sugar syrup to reach a final concentration of 300, 3000 and 6000 ppm. The final concentration of EtOH in the diets was 1.6%, which is below the reported value for acute exposures, with no effects on behavioural components [33]. To test if the solvent generated alterations *per se* in the response variables, a group of control honeybees [hereafter referred to as "Control (ethanol)"] received the same amount of ethanol in the syrup. Control groups of honeybees ("Control") fed just on syrup without ethanol were also included.

**Experiments performed.** Two experiments were run to assess the effect of the EOEb ingestion and health status on the recorded variables:

1. Experiment I tested the ingestion by non-infected honeybees (hereafter referred to "healthy" honeybees; although the complete health status is not known) of the two EOEb tested at different doses (300, 3000 y 6000 ppm) provided in the diet (5 replicates per treatment, including the corresponding controls). Treatments in this experiment will be hereafter referred to: EOEb_2009–300; EOEb_2009–3000; EOEb_2009–6000; EOEb_2014–300; EOEb_2014– 3000; EOEb_2014–6000; Control and Control (ethanol). The experiment was run for 12 days (until honeybees were 14-days old).

2. Experiment II tested the effect of ingestion by healthy and *Nosema*-infected bees of the two EOEb at 3000 ppm (5 replicates per treatment, including the corresponding controls).

Treatments in this experiment will be then referred to as: EOEb_2009; EOEb_2009+Nos; EOEb_2014; EOEb_2014+Nos; Control; Control+Nos; Control (ethanol); and Control (ethanol)+Nos (where "+Nos" stands for honeybees previously infected with *Nosema cera-nae*). The feeding on diets was performed for 8 days, until honeybees were 10-days old, after which they were used to extract the midgut and perform the CHC extraction. This experiment was shorter than experiment I due to the mortality of *Nosema*-infected bees.

To obtain non-infected and *Nosema*-infected workers for this experiment, two-day-old honeybees were individually feed [34] with 10 μL of 0% sugar solution (healthy workers) or with a mixture including freshly extracted spores of *N. ceranae* at a concentration of 1 x $10^5$ spores/μL (*Nosema*-infected workers). The infection with *N. ceranae* was achieved as detailed previously [35], and confirmed by PCR analysis at the end of the assays [36].

**Assessed variables.** *Survival and diet consumption*. Honeybees without movement response after mechanical stimulation were daily recorded as dead and removed from containers. In addition, gravity feeders were weighted every day to estimate consumed amounts. To correct consumed volumes for diet evaporation, identical feeding devices without honeybees were similarly weighed.

Nosema ceranae *development*. In Experiment II, 10 days post-infection, fifteen to twenty honeybees per replicate were analyzed to individually quantify the parasite development. The digestive tracts of the honeybees were dissected and the midguts were separated and stored at 20 ˚C until spore quantification. The number of spores was individually estimated with a hemocytometer under a light microscope [37].

*CHC quantification*. Extracts were obtained from pooled honeybees (5 bees/replicate) to detect and quantify less abundant CHC such as methyl-branched alkanes or alkadienes. The bees were sacrificed and whole-body extracts were prepared in hexane (1.6 mL) [38] with the addition of 200 μL of internal standard solution (tridecane at 50 ng/μL). The extract was concentrated under a nitrogen stream to 200 μL, and 1 μl was analyzed by gas chromatography coupled to mass spectrometry (GCMS).

## Chemical characterization of essential oils and cuticular hydrocarbons

The identification of individual compounds in the EOEb and CHC extracts was carried out using a Shimadzu 2010 Gas Chromatographer coupled to a Shimadzu QP2010 plus mass spectrometer (MS). Data were analyzed using the Shimadzu Corporation GC-MS Solution v2.50 software (1999–2006). Analyses were run on a DB5-MS column (30 m x 0.25 mm id, 0.25 μm film thickness) provided by Macherey–Nagel (Düren, Germany). The carrier gas was helium at 1 mL/min. The injection volume was 1 μL. MS parameters were: electron impact ionization at 70 eV ionization potential (scan mode), m/z 40–550. The injector temperature was 250˚C and the interphase temperature was 300˚C. For EOEb characterization, the oven temperature program was as follows: 40˚C for 2 min, increase to 240˚C at 5˚C/min, 240˚C for 1 min, the-nincrease to 320˚C at 10˚C/min. Injections were performed in split mode (30:1). For CHC characterization, the temperature of the GC oven was programmed from an initial temperature of 70˚C (1 min), then heated to 300˚C at 5˚C/min and held for 1 min. Injections were performed in splitless mode. Chemical characterization was performed by comparison of mass spectra and arithmetic retention indexes [39] to those reported in the NIST2008 [40], SHIM2205 [39] and Pherobase [41] databases, as well as in the literature [38] and with synthetic standards when available. Therefore, the identification of the components was done to the level I and II of the Metabolomics initiatives [42].

In the case of the EOEb, a relative quantification was done by normalization of the total area. In the case of CHC, quantification was done based on the relative areas of compounds compared to the area of the internal standard.

## Statistical analyses

The composition of the EOEb extracted from the different plant materials, at different extraction times were compared by χ2 contingency analyses (VassarStats website for statistical computation [43]).

In each experiment, comparisons between controls and controls with ethanol were done for mortality using Log-Rank tests and for food consumption using Mann-Whitney tests. Infection levels with *Nosema* spores were also compared by Mann-Whitney tests. CHC contents were evaluated individually and familywise (group of compounds). Amounts were compared after checking for normality (Anderson-Darling procedure) by ANOVA (GLM) using diet, EOEb doses and health status as factors where corresponded; and post-ANOVA pairwise multiple comparison procedures were performed by Tukey simultaneous tests. CHC profiles were also analyzed by multivariate analyses (by exploratory Principal component analyses first and then Partial Least Squares (PLS)- Discriminant Analysis (DA) on normalized, scaled, and centered data. PLS-DA were validated by permutation tests [44]. Statistical analyses were run using MINITAB® 17.3.1 and the Metaboanalyst platform [44].

## Results

### Essential oil composition

The complete composition of the EOs used in the bioassays was reported elsewhere [6]. Considering the main compound classes, the composition of both EOEb did not differ between the two plant collecting years (Chi-Square, Cramer's V tests: χ2 = 0.81, df = 3, P = 0.85). The pooled composition consisted of monoterpene hydrocarbons (48 ± 1%), oxygenated monoterpenes (0.4 ± 0.2%), sesquiterpene hydrocarbons (42 ± 4%) and oxygenated sesquiterpenes (1.8 ± 0.8%). The main compounds (> 5%) were β-caryophyllene (5 ± 2%), limonene (5.2 ± 0.8%), β-elemene (6.6 ± 0.1%), sabinene (7 ± 1%), β-pinene (7 ± 1%), germacrene D (7.7 ± 0.13%), (E)-β-guaiene (9 ± 2%) and α-pinene (18 ± 5%). Even though the compositions of both EOEb were similar, some slight differences in individual compounds were detected (the main differences between both EOEb were found in α-pinene (22 vs 15%), sabinene (6 vs 8%), β-pinene (6 vs 8%), β-caryophyllene (6 vs 4%) and (E)-β-guaiene (10% vs 7% in EOEb_2009 and EOEb_2014, respectively). Since it is well known [7] that minor differences in EO composition can cause changes in bioactivity due to synergistic or antagonistic effects, both EOEb were tested separately on honeybees.

### *Nosema ceranae* development

As previously mentioned, experiments were run with "healthy" (non-infected by *Nosema*) and *Nosema*-infected honeybees. At the end of the experiments, no spores were found in midguts from non-infected honeybees. Midguts from infected honeybees fed with the EOEb and infected controls exhibited similar spore counts (Kruskal-Wallis test; H Statistic = 5.6, df = 4; p = 0.23; overall mean: $(2 ± 2) \times 10^6$ spores/bee), indicating an absence of activity of the EOEb on midgut spores.

## Survival and food consumption

**Ethanol effect on survival and food consumption.** In none of the experiments the consumption of ethanol affected the survival (Log-Rank Tests: Experiment I: Statistic: 0.0898, df = 1, p = 0.764; Experiment II: Statistic: 0.589, df = 1, p = 0.443), or the food consumption (Mann-Whitney tests: Experiment I: U Statistic = 7.000, T = 19.000, p = 0.886; Experiment II, U Statistic = 22.000, T = 18.000, p = 0.06), in comparison with the control without ethanol.

**Effect of EOEb intake on survival and food consumption.** When studying the effect of EOEb ingestion on healthy bees (Experiment I), food consumption at the end of the assay was not different for honeybees fed with either of the EOs at different doses (300 to 6000 ppm) and the control honeybees (ANOVA, GLM, p = 0.46 for diet and P = 0.20 for dose, Fig 1A); neither was the survival among groups (ANOVA, GLM, p = 0.75 for treatment, p < 0.05 for time, Fig 2A). Cumulative mortality reached an average of 5 ± 3% at the end of the assays for all treatments.

In experiment II, in which the effect of EOEb consumption was studied in non-infected and infected bees, cumulative food consumption was not affected by the ingestion of EOs (ANOVA GLM, P = 0.49 for diet and P = 0.30 for the interaction diet*health status, Fig 1B). Interestingly, *Nosema*-infected honeybees showed a significant increase in food uptake

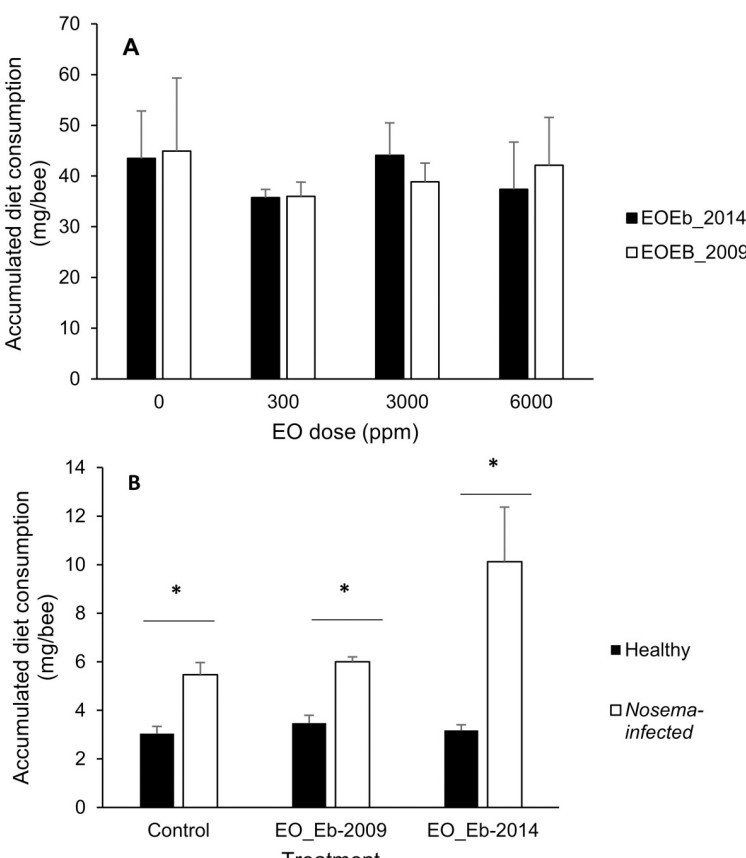

**Fig 1. Cumulative food consumption by honeybees fed on diets enriched with *E. buniifolium* EO.** EOs from different harvest times (2009 and 2014) were tested: on healthy honeybees at different doses (**A**) and on healthy and *Nosema*-infected honeybees at 6000 ppm (**B**). No differences were found among consumption on different diets by healthy bees (ANOVA, GLM, p > 0.05, **A**). *indicates significant differences between bar pairs (ANOVA, GLM, p < 0.01; **B**).

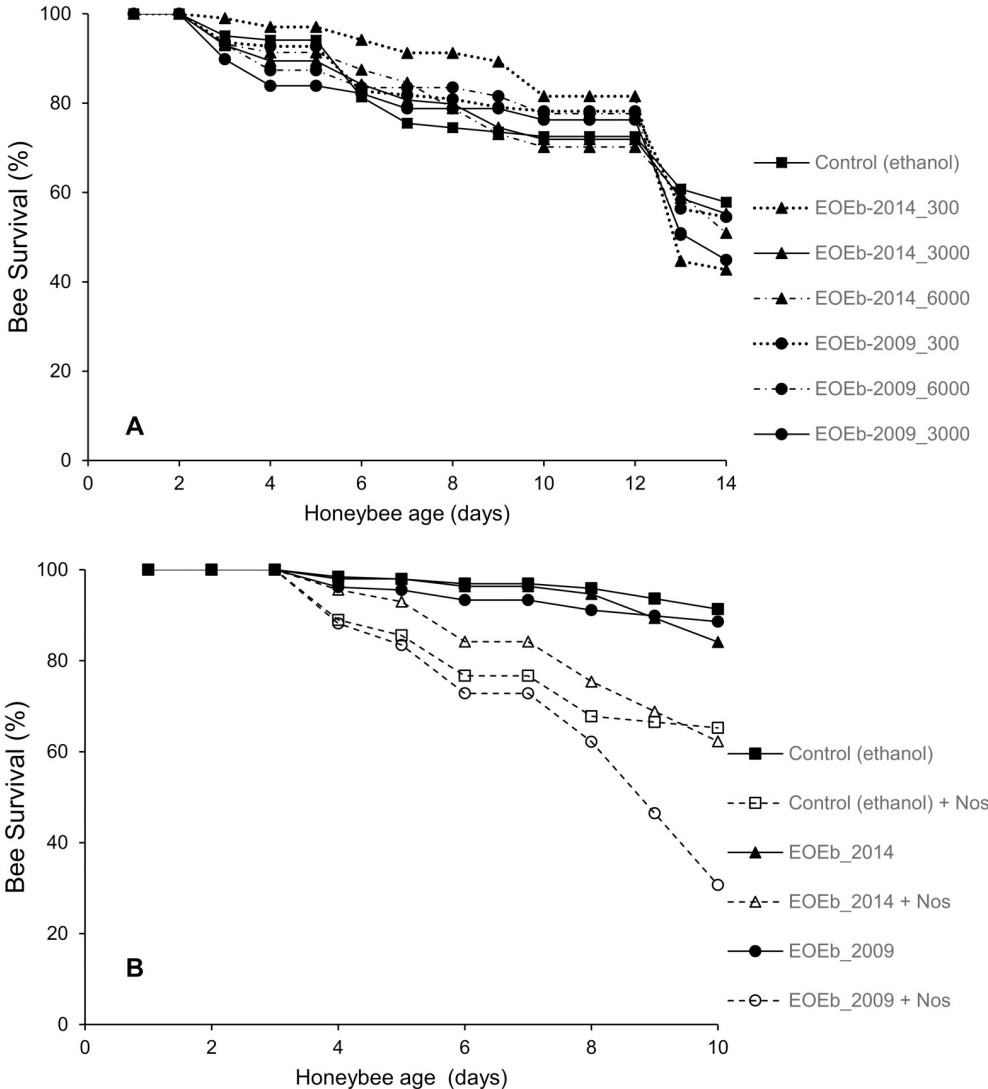

**Fig 2. Survival of honeybees fed on diets enriched with *E. buniifolium* EO.** EOs produced at different harvest times (2009 and 2014) were tested: on healthy honeybees at different doses (**A**) and on healthy (solid symbols) and *Nosema*-infected (open symbols) honeybees at 3000 ppm (**B**). For healthy bees (**A**), no differences were found among honeybees fed with the EOEb at different doses (ANOVA, GLM, p = 0.65 for treatment and p = 0.13 for the interaction treatment*time), and survival decreased with time (p < 0.05). For *Nosema*-infected honeybees (**B**), significant differences (ANOVA; GLM) were found related to health status (P < 0.001) and time (P < 0.001) but not related to diet (P = 0.6). Error bars are not shown for clearness of the figure.

(ANOVA GLM, P < 0.01, Fig 1B). As expected [45], survival of honeybees was affected by the infection with *Nosema* (ANOVA, GLM p < 0.001), and decreased with time (p < 0.001), but not by the consumption of EOEb (p = 0.6, Fig 2B), in line with the results obtained in experiment I.

## Effect of diets on CHC

**Overall results and effect of ethanol consumption on CHC.** CHC data were analyzed by identifying and quantifying -relative to the internal standard area- 39 CHC (S1 and S2 Tables).

Among these compounds, 4 were not identified and the rest belonged to one of the 4 following compound groups: linear alkanes (hereafter referred to alkanes), alkenes, alkadienes and branched alkanes [38] (S1 and S2 Tables and S1 Fig). The major compounds were the linear alkanes n-pentacosane, n-heptacosane and n-tricosane, accounting all together for c.a. 67% of the CHC in both experiments (S1 and S2 Tables). Among branched alkanes, the methylhetpta-cosanes were the most abundant in both experiments, although in lower amounts (c.a. 1%, S1 and S2 Tables). These results are similar to many previous studies (see [46, 47], among others). In the case of non-saturated hydrocarbons, major compounds differ between experiments. The alkenes 9-pentacosene, 9-heptacosene and 9-tricosene, and the alkadiene hentriaconta-diene, were the main unsaturated CHC in Experiment I. In Experiment II, however, tritriacon-tene, 9-pentacosene, 9-hentriacontene were the main monounsaturated alkenes, while nonacosadiene and pentacosadiene were the main alkadienes. These results are not surprising, since both experiments were run on bees from different colonies that can differ in their non-saturated CHC profile, a fact that is well known to be related to nest recognition [48].

Since the EOEb was added to the diet in ethanol solution, we first checked the effect of etha-nol ingestion on CHC production: when comparing CHC from honeybees fed with syrup with those from honeybees fed with syrup and ethanol, there was no grouping pattern in a Principal Component Analysis (PCA) according to diet, and none of the supervised analyses done after-wards generated a valid model (permutation tests fails for either PLSDA and ortho-PLSDA), indicating that no differences between both sample types could be detected. Moreover, ANOVA on compound groups (Fig 3) and t-tests on either the 39 identified CHC or the com-pound groups also showed no significant differences (p > 0.05 in all cases. See S3 Table for t-tests on the 39 compounds, for Control vs Control (ethanol) samples in both experiments). Therefore, ethanol consumption (at 1.6%) did not affect CHC profiles.

**Effect of EOEb ingestion at different doses on CHC of healthy bees (Experiment I).** The consumption of either of the EOEb (from different harvest times) did not affect the four groups of CHC analyzed (Table 1, ANOVA, GLM p > 0.05). However, ingestion of the highest concentration of EOEb had a significant effect on the CHC compound groups (Table 1 and Fig 3). Significantly increased CHC production was observed with a diet concentration of 6000 ppm for both EOEb (Fig 3).

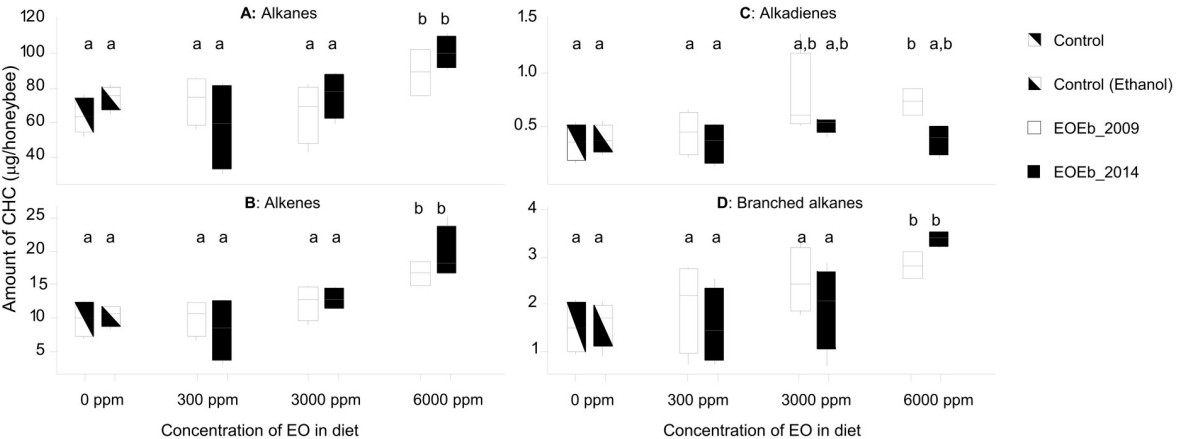

**Fig 3. Boxplot of compound groups of CHC from healthy honeybees (Experiment I): Alkanes (A), Alkenes (B), Alkadienes (C) and Branched alkanes (D).** CHC were extracted from honeybees fed on EO-enriched diets (2 EOEb harvested at different times) with different EOEb concentrations. Means with different letters are significantly different in CHC content (within each graph, ANOVA, GLM, with post-test Tukey comparisons, p < 0.05, in all cases; see Table 1 for details).

**Table 1. Results on the analysis of variance (GLM) for groups of CHC from healthy honeybees fed on 2 EOEb at different doses (0, 300, 3000, 6000 ppm; Experiment I).**

|  | Alkanes | | | Alkenes | | |
|---|---|---|---|---|---|---|
|  | **Dose** | **Product (EOEb)** | **Dose*Product** | **Dose** | **Product (EOEb)** | **Dose*Product** |
| DF | 3 | 1 | 3 | 3 | 1 | 3 |
| Adj SS | 81263178 | 1336926 | 14235539 | 301.512 | 0.131 | 21.639 |
| Adj MS | 27087726 | 1336926 | 4745180 | 100.504 | 0.131 | 7.213 |
| F-Value | 5.99 | 0.3 | 1.05 | 11.21 | 0.01 | 0.8 |
| P-Value | **0.004** | 0.592 | 0.39 | **< 0.001** | 0.905 | 0.505 |
|  | Alkadienes | | | Branched Alkanes | | |
|  | **Dose** | **Product (EOEb)** | **Dose*Product** | **Dose** | **Product (EOEb)** | **Dose*Product** |
| DF | 3 | 1 | 3 | 3 | 1 | 3 |
| Adj SS | 0.19917 | 0.07595 | 0.0797 | 189.08 | 0.001 | 33.279 |
| Adj MS | 0.06639 | 0.07595 | 0.02657 | 63.0265 | 0.0014 | 11.0929 |
| F-Value | 3.33 | 3.81 | 1.33 | 9.21 | 0 | 1.62 |
| P-Value | **0.038** | 0.064 | 0.289 | **< 0.001** | 0.989 | 0.213 |

Significant differences were found related to the doses (p-values in bold) of EOEb but not to the product (EO harvested at different times) or the interaction product*dose.

To assess individual compounds that may account for the differences found in compound groups (Fig 3), and since the EOEb harvested at different years were not different regarding CHC groups, data from the two EOEb were pooled and the whole set was re-analyzed. A multivariate analysis run on the complete set of variables allowed to differentiate samples coming from honeybees fed on diets enriched with EOEb at 6000 ppm from all others (Fig 4A): the general trend being an increase in CHC in parallel with an increase in EOEb intake. When all CHC compounds were compared among samples, 17 of them (Table 2) were significantly different in samples from honeybees fed on diets with EOEb at 6000 ppm (ANOVA, p < 0.05 in all cases, see Table 2 for the detailed analyses), and all of them were up-regulated, with the

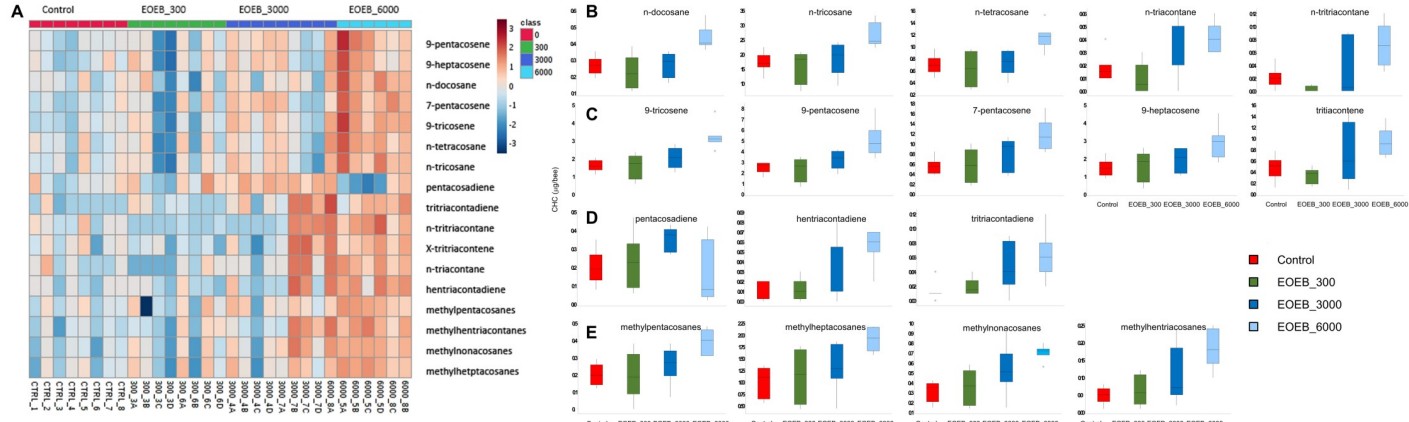

**Fig 4.** Agglomerative, hierarchical cluster analysis on the individual CHC in honeybees fed on EO-enriched diets at different doses: Heatmap (**A**, distance measured using Euclidean, and clustering algorithm using ward.D in Metaboanalyst [44]). The colored boxes on the right indicate the relative concentrations of the corresponding metabolite in each group under study. Boxplots of the selected CHC that accounted for significant differences (identified from variable importance in projection (VIP) in a PLS-DA: **B**, linear alkanes; **C**, Alkenes, **D**, Alkadienes and **E**, branched alkanes. Boxes represent 95%-intervals around the media (horizontal bar within a box) calculated as 1.58*IQRT /sqrt(n).

**Table 2. Results for individual CHC in experiment I (honeybees were fed on EOEb at different doses: 0, 300, 3000, 6000 ppm) where significant differences were found related to the doses by the analysis of variance (ANOVA, GLM) and variable importance in projection (VIP scores are reported when > 1) from a PLS-DA.**

| CHC Compound | Formula | ANOVA | | | PLS-DA |
|---|---|---|---|---|---|
| | | f.value | p | Post ANOVA test* | VIP |
| **Alkanes** | | | | | |
| n-docosane | $C_{22}H_{46}$ | 6.86 | 0.001 | 6000–0; 6000–300; 6000–3000 | 1.11 (C1$^\xi$); 0.98 (C2); 0.96 (C3); 0.9 (C4); 0.89 (C5) |
| n-tricosane | $C_{23}H_{48}$ | 4.96 | 0.007 | 6000–0; 6000–300; 6000–3000 | 1.08 (C1); 0.83 (C2); 0.85 (C3); 0.81 (C4); 0.79 (C5) |
| n-tetracosane | $C_{24}H_{50}$ | 5.12 | 0.006 | 6000–0; 6000–300; 6000–3000 | 1.07 (C1); 0.91 (C2); 0.92 (C3); 0.87 (C4); 0.87 (C5) |
| n-triacontane | $C_{30}H_{62}$ | 6.27 | 0.002 | 6000–0; 3000–300; 6000–300 | 1.19 (C1); 0.9 (C2); 0.87 (C3); 0.88 (C4); 0.87 (C5) |
| n-tritriacontane | $C_{33}H_{68}$ | 6.96 | 0.001 | 6000–0; 3000–300; 6000–300; 6000–3000 | 1.08 (C1); 0.87 (C2); 0.85 (C3); 0.99 (C4); 1.05 (C5) |
| **Alkenes** | | | | | |
| 9-tricosene | $C_{23}H_{46}$ | 9.06 | 0.0003 | 6000–0; 3000–300; 6000–300; 6000–3000 | 1.42 (C1); 1.14 (C2); 1.12 (C3); 1.08 (C4); 1.05 (C5) |
| 9-pentacosene | $C_{25}H_{50}$ | 7.6 | 0.001 | 6000–0; 3000–300; 6000–300; 6000–3000 | 1.31 (C1); 1.04 (C2); 1.03 (C3); 1 (C4); 0.97 (C5) |
| 7-pentacosene | $C_{25}H_{50}$ | 7.3 | 0.001 | 6000–0; 6000–300; 6000–3000 | 1.35 (C1); 1.22 (C2); 1.19 (C3); 1.26 (C4); 1.25 (C5) |
| 9-heptacosene | $C_{27}H_{54}$ | 3.68 | 0.02 | 6000–0; 6000–300 | 1.12 (C1); 0.85 (C2); 0.85 (C3); 0.88 (C4); 0.85 (C5) |
| tritriacontene | $C_{33}H_{66}$ | 3.76 | 0.022 | 6000–0; 3000–300; 6000–300 | < 1 (C1 to C5) |
| **Alkadienes** | | | | | |
| pentacosadiene | $C_{25}H_{48}$ | 5.93 | 0.003 | 3000–0; 3000–300; 3000–6000 | < 1 (C1 to C4); 1.06 (C5) |
| hentriacontadiene | $C_{31}H_{60}$ | 5.52 | 0.004 | 6000–0; 6000–300; 6000–3000 | 1.26 (C1); 0.95 (C2); 0.99 (C3); 0.95 (C4); 0.93 (C5) |
| tritriacontadiene | $C_{33}H_{64}$ | 7.77 | 0.001 | 3000–0; 6000–0; 3000–300; 6000–300 | 1.46 (C1); 1.7 (C2); 1.68 (C3); 1.58 (C4); 1.54 (C5) |
| **Branched alkanes** | | | | | |
| methylpentacosanes | C26H54 | 4.06 | 0.017 | 6000–0; 6000–300 | 1.05 (C1); 0.83 (C2); 0.81 (C3); 0.78 (C4); 0.78 (C5) |
| methylheptacosanes | C28H58 | 4.48 | 0.011 | 6000–0; 6000–300 | 1.23 (C1); 0.95 (C2); 0.93 (C3); 0.87 (C4); 0.86 (C5) |
| methylnonacosanes | C30H62 | 6.47 | 0.002 | 3000–0; 6000–0; 3000–300; 6000–300 | 1.42 (C1); 1.1 (C2); 1.09 (C3); 1.02 (C4); 1.04 (C5) |
| methylhentriacontanes | C32H66 | 6.07 | 0.003 | 3000–0; 6000–0; 6000–300 | 1.4 (C1); 1.14 (C2); 1.11 (C3); 1.06 (C4); 1.05 (C5) |

Model validation by permutation tests based on separation distance; p value based on permutation 0.0085 (17/2000).

* treatments with a dash are significantly different (post-ANOVA pairwise multiple comparisons, Tukey simultaneous tests).

$^\xi$ Ci stands for component I in the PLS-DA.

exception of pentacosadiene (Fig 4B–4E). Similar results were found when a model from a PLS-DA was built ($R^2$ = 0.9 and $Q^2$ = 0.5 for 5 components, the model passed permutation test with p = 0.0085). From that model, the compounds with a Variable Importance in Projection (VIP) > 1 [44] were also identified (Table 2). This approach coincidently identified 16 out of the 17 previously discovered in the conventional analyses (Table 2) with the exception of tri-triacontene that had a VIP score < 1 for all components (0.98 for component C1; 0.78, C2; 0.77, C3; 0.86, C4; 0.83, C5).

In summary, in healthy honeybees fed with EOEb-enriched diets there was an increase of CHC for all CHC groups (Table 3) related to the highest doses of EOEb included in the diet (6000 ppm). Among the main alkanes (n-pentacosane, n-heptacosane and n-tricosane) only n-tricosane was affected. Moreover, all the main alkenes (9-pentacosene, 9-heptacosene and 9-tricosene) and branched alkanes were also up-regulated. Alkadienes in turn were the less affected CHC.

**Effect of EOEb consumption on CHC from Nosema-infected honeybees (Experiment II).** When the effect of health status (*Nosema*-infected vs healthy honeybees) combined with EOEb ingestion was studied (experiment II), no differences were found in the CHC groups among samples (Fig 5; ANOVA GLM p > 0.05 in all cases, see Table 4 for detailed analyses).

**Table 3. Overall amounts of CHC from healthy honeybees (experiment I) fed on EOEb at different doses.**

|  | Control | EOEb_300 | EOEb_3000 | EOEb_6000 |
|---|---|---|---|---|
| Alkanes | 69 ± 4[a] | 65 ± 7[a] | 71 ± 5[a] | 96 ± 5[b] |
| Alkenes | 10 ± 1[a] | 9 ± 1[a] | 13 ± 1[a] | 18 ± 1[b] |
| Alkadienes | 0.37 ± 0.05[a] | 0.40 ± 0.07[a] | 0.6 ± 0.1[b] | 0.5 ± 0.1[a,b] |
| Branched Alkanes | 1.5 ± 0.2[a] | 1.7 ± 0.3[a] | 2.2 ± 0.3[a,b] | 3.2 ± 0.1[b] |
| All CHC[a] | 83 ± 4[a] | 80 ± 9[a] | 91 ± 7[a] | 126 ± 7[b] |

Pooled data from both EOEb tested are shown (see text). Different letters within each row indicate significant differences (ANOVA-GLM, with post-test Tukey comparisons, $p < 0.05$ in all cases).

Comparing the amounts of individual compounds (S2 Table), no significant differences among treatments were found for none of the compounds as a function of diet for either of the EOEb (ANOVA, GLM $p > 0.05$ in all cases). This result indicates that the consumption of EOEb at 3000 ppm did not affect the CHC levels in *Nosema*-infected bees, as it was also found in experiment I for healthy honeybees.

On the other hand, when comparing CHC profiles from healthy and *Nosema*-infected bees, slight but significant differences were found on some individual compounds in relation to health status, namely n-nonadecane ($p = 0.021$), n-heneicosane ($p = 0.004$), 7-pentacosene ($p = 0.024$), n-pentacosane ($p = 0.014$) and n-heptacosane ($p = 0.048$, ANOVA, 2 factors in all cases, see S4 Table). Whereas n-nonacosane and n-heneicosane were down-regulated in *Nosema*-infected bees, the other compounds were up-regulated (Fig 6). These findings were consistent with a discriminant analysis (PLS-DA, number of components needed: 5, performance: $R^2 = 0.81$, $Q^2 = 0.32$, permutation test $P = 0.002$) run on the overall chemical profiles, considering only the factor of *Nosema* infection (Fig 7). In this PLS-DA, other compounds were also detected to be up-regulated as indicated by the scores of VIP (Fig 7B). Our results are in line with previous reports [49] in which differences in CHC profiles were detected between honeybees infected with *Nosema* and non-infected.

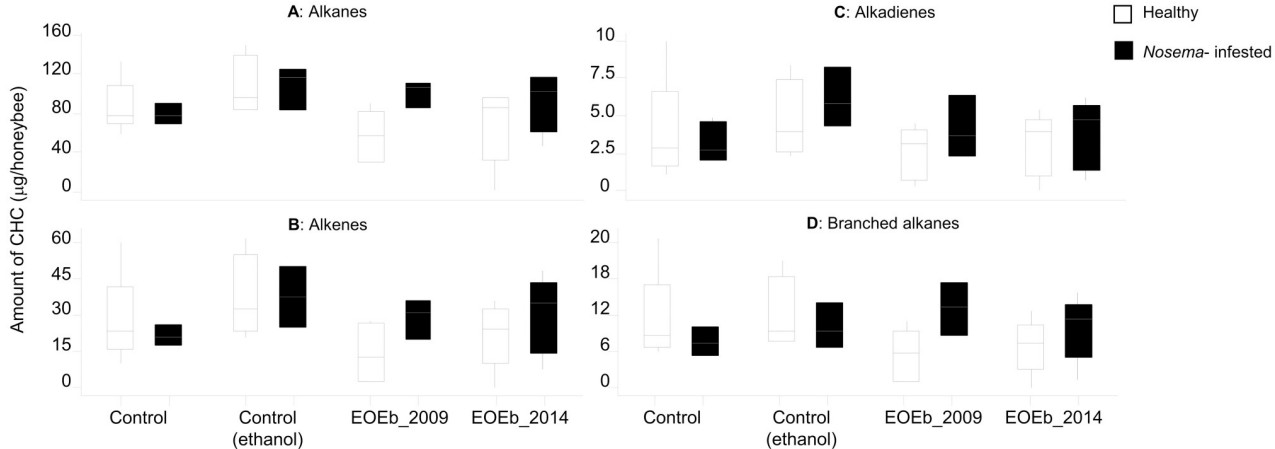

**Fig 5. Boxplot of compound groups of CHC from honeybees with different health status (Experiment II): Alkanes (A), Alkenes (B), Alkadienes (C) and Branched alkanes (D).** CHC were extracted from honeybees fed on EO-enriched diets (2 EOEb harvested at different times, applied at 3000 ppm) with different health status. No significant differences were found (ANOVA, GLM, with post-test Tukey comparisons, $p > 0.05$ in all cases, see Table 2) for either of the CHC groups between healthy and *Nosema*-infected honeybees.

**Table 4. Results on the analysis of variance (GLM) for groups of CHC from honeybees with different health status (healthy and *Nosema*-infected honeybees in Experiment II) fed on two EOEb harvested at different times (product) at 3000 ppm.**

| | Alkanes | | | Alkenes | | |
|---|---|---|---|---|---|---|
| | Health status | Product (EO) | H. status * Product | Health status | Product (EO) | H. status * Product |
| DF | 1 | 3 | 3 | 1 | 3 | 3 |
| Adj SS | 2014 | 4021 | 3291 | 152.1 | 1031 | 528 |
| Adj MS | 2014 | 1340 | 1097 | 152.1 | 343.6 | 176 |
| F-Value | 2.49 | 1.66 | 1.36 | 0.75 | 1.7 | 0.87 |
| P-Value | 0.13 | 0.20 | 0.28 | 0.393 | 0.19 | 0.47 |
| | Alkadienes | | | Branched Alkanes | | |
| DF | 1 | 3 | 3 | 1 | 3 | 3 |
| Adj SS | 4.9 | 20.92 | 7.49 | 14.6 | 29.3 | 152.7 |
| Adj MS | 4.9 | 6.97 | 2.50 | 14.6 | 9.8 | 50.9 |
| F-Value | 0.88 | 1.26 | 0.45 | 0.53 | 0.36 | 1.86 |
| P-Value | 0.36 | 0.36 | 0.72 | 0.47 | 0.79 | 0.16 |

No significant differences were found related to either of the factors or the interaction product*health status.

## Discussion

The long-term sublethal effects of xenobiotics on honeybees [10, 14], and specifically of acaricides used in beehives, have been somewhat studied [13]. Yet, information is scarce in particular for essential oils or their individual components [27, 50–52]. Sub-lethal effects on adult honeybees include changes in their appetitive behavior [18, 20], motor function, grooming and wing fanning behaviors [53], olfactory memory and learning performance [20–23, 51], and reduction of phototaxis [27]. Terpenoids usually found in EO (such as thymol and carvacrol) have been recently reported to increase activity of acetylcholinesterase at sub-lethal doses [50], possibly causing motor-related effects. In this study, the effect of chronic consumption of the acaricide [5] essential oil from *E. buniifolium* on survival and food consumption was evaluated. EOEb was applied in the diet to ensure that the putative sub-lethal effects could be

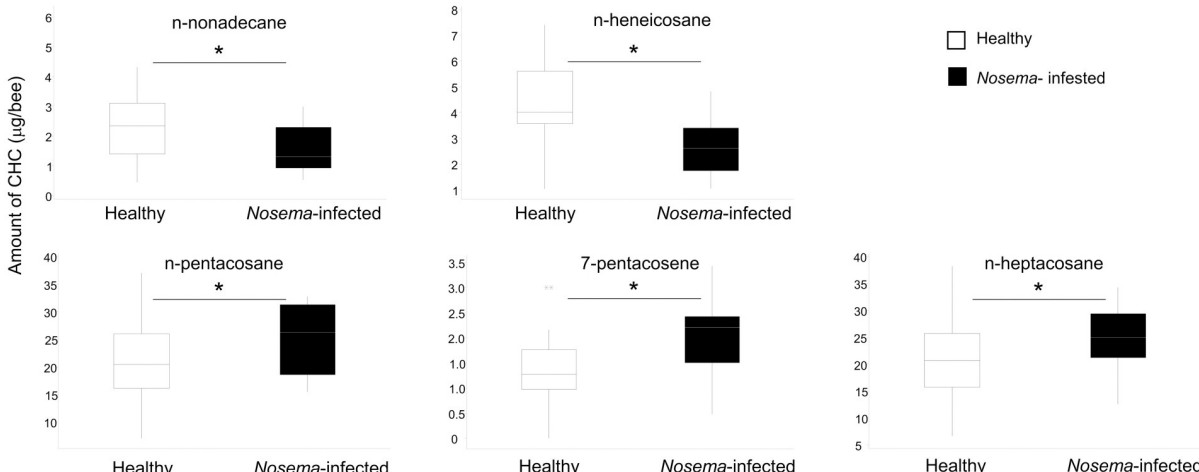

**Fig 6. Boxplot of the 5 compounds where significant differences were found between healthy and *Nosema*-infected honeybees.**
ANOVA-GLM, 2-factors, P > 0.05 for diet and P < 0.05 for health status for the 5 compounds, see S2–S4 Tables for the complete statistical analyses.

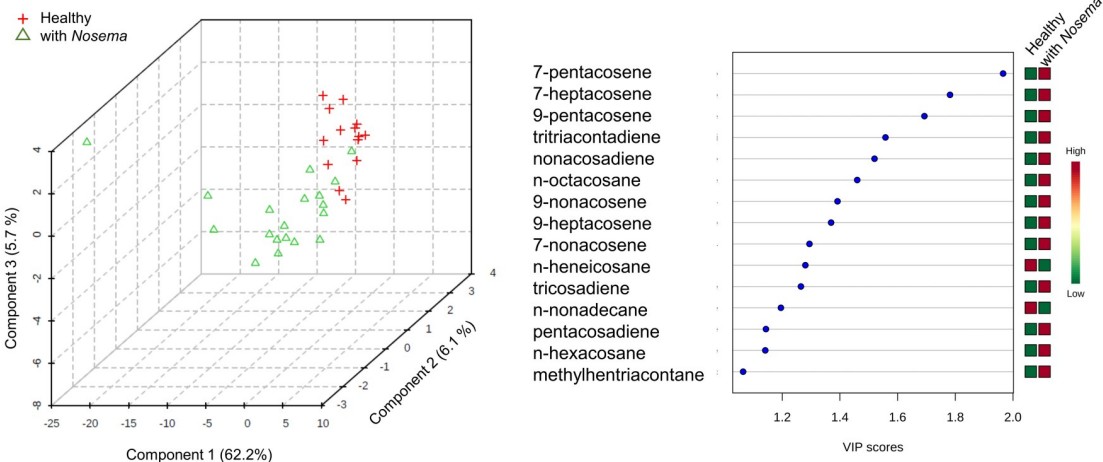

**Fig 7.** Multivariate analysis of CHC from healthy and *Nosema*-infected honeybees: 3-D scores plot (**A**) between selected components corresponding to a PLAS-DA run on samples form healthy and *Nosema*–infected bees. The explained variances are shown in brackets. The model was validated (number of components needed, 5; performance: R2 = 0.81; Q2 = 0.32, permutation test p = 0.002). Important features (VIP > 1) found by PLS-DA (**B**): colored boxes on the right indicate the relative concentrations of the corresponding metabolite in each group under study.

observed, since the toxicity of xenobiotics in honeybees is usually greater by oral intake [54]. Moreover, direct feeding with sanitary products provides a more direct and systemic approach for the exposure of the etiological agents of various honeybee pathologies, and circumvents issues related to variable fumigant volatility [52]. Neither survival nor the amount of diet consumed by honeybees was affected by the consumption of EOEb (up to 6000 ppm in the diet). Similar results have been previously reported for honeybees exposed to sublethal doses of imidacloprid [18] and of carvacrol, an ubiquitous EO component [50]. In the case of thymol [50], food intake did vary with increasing doses; and for glyphosate, food intake varied depending on the season when glyphosate was applied [20]. We did not evaluate the effects of EOEb consumption on different seasons or environmental conditions, so a similar seasonal modulation of EOEb effects on food uptake or survival cannot be discarded.

The intake of the EOEb at the lower doses tested (300 to 3000 ppm) did not cause changes in the CHC amounts or their profiles. In our experiments the EO was incorporated into the diets as an ethanol solution, and we here showed that ethanol consumption (at 1.6%) did not affect CHC profiles either. Although this is a side result concerning our working hypothesis, it is nonetheless ecologically relevant since forager honeybees are usually exposed to different degrees of ethanol from fermented nectars during their foraging activities, and they transport ethanol to the hive where nurses are exposed [55]. If exposure to ethanol were to modify CHC profiles, one of the chemical cues for nestmate discrimination would be compromised [56].

CHC cover the outer surface of insects not only playing a structural role, providing protection against desiccation, but also acting as pheromones [57]. In honeybees, CHC have been characterized in many studies as a mixture of linear and branched alkanes, alkenes, and alkadienes (reviewed by Blomquist, 2010 [58]). Among these classes, whereas alkanes play an important role in preventing desiccation, the other groups are important as part of the recognition system among nestmates [48], as it is the case in many other hymenopterans [56]. According to our findings, all CHC groups can be affected when the diet includes a concentration of 6000 ppm of EOEb, potentially affecting all postulated CHC functions. Specifically, the modification of alkenes, alkadienes and branched alkanes (Table 2 and Fig 4) may modify the

chemical signature and lead to a failure in nestmate recognition among members of the colony [56, 58, 59]. This effect may be more pronounced in older bees that have higher olfactory sensitivity toward CHC odors, and also higher frequencies of interactions with nestmates [60]. In fact, the CHC shown in this study to change with EOEb intake include 4 short chain (< 29C) alkenes (9-tricosene, 9-pentacosene, 7-pentacosene, 9-heptacosene), previously identified as more relevant cues in social recognition than other CHC, because they are better discriminated by adult honeybees [61, 62].

CHC are internally biosynthesized and then transported to the cuticular surface [57]. Indeed, internal hydrocarbon profiles are qualitatively similar to CHC in honeybees for alkenes and alkadienes [62, 63]. The CHC biosynthetic pathways in honeybees are not only dependent on age but also sensitive to environmental cues [2, 64]. While it has been reported that hemolymph hydrocarbons can also be affected by the incorporation of exogenous dietary hydrocarbons [63], these changes do not affect CHC composition. Therefore, the effect in CHC in honeybees fed with EOEb at 6000 ppm may point to a direct effect of the ingested EO on the biosynthetic pathway of CHC, an hypothesis that deserves further research.

Our results show that nurse honeybees that have been fed with high doses of EOEb-enriched diets produce CHC profiles with higher amounts of 9-tricosene, 9-pentacosene and tricosane (Table 2 and Fig 4). Waggle-dancing honeybees increase the emission of Z-(9)-tricosene, Z-(9)-pentacosene, tricosane and pentacosane, reportedly as part of the recruitment process towards a food source [47]. Is it hence conceivable that the increase of three of these four CHC compounds found in our study could disrupt the process of worker recruitment, especially in 14-day-old nurse honeybees that are at the onset of their transition to foraging [65].

Exogeneous stressors have also been proved to change CHC profiles. For instance, modifications of the cuticular hydrocarbons by *Varroa* infection have been reported [49, 59, 66]. In these studies, infestation affectes CHC in their total amounts [66] and relative concentrations [59]. *Nosema*-infected bees also exhibited different CHC profiles compared to healthy bees [49, 67], although not as clearly as in the case of varroa. In line with these reports, we here found changes in CHC correlated with the *Nosema* infection (Fig 7). As an additional contribution from our results, we found that CHC profiles of *Nosema*-infected honeybees were not differentially affected by the consumption of the EOEb. Besides, although some EO or their components when ingested reduce the counts of *Nosema* spores [68–70], we have here shown that consumption of EOEb was not successful in lowering *Nosema* spores as it was previously reported for other EO [28].

CHC were assessed due to their key role as pheromones involved in nestmate recognition, as well as in their proposed communicative role during the waggle dance and in the maintenance of the homeostasis of the colony [3, 47, 57]. Though which sub-lethal effects to assess has been a matter of controversy [71], the key role played by CHC in communication among adult honeybees makes them good potential indicators of disruptive effects caused by chemical sanitary treatments, as well as chronic effects resulting from organic products used in beekeeping. Our results show that the ingestion of an EO can affect, as other xenobiotics do, the CHC profiles of honeybees, and that the effect is dose-dependent. The overall effect was an increase in the amounts of CHC when the bees were fed with high doses of EOEb-enriched diet (Table 2 and Fig 4). Oral administration up to 3000 ppm of EOEb appears to be safe for honeybees, therefore this amount would indicate the maximum safe application dose for a botanical pesticide to be developed based on this essential oil. So we chose this concentration to be tested on *Nosema*-infected honeybees, and likewise healthy honeybees we found no effect on the CHC in *Nosema*-infected bees.

Our results underline that the development of organic pesticides to be used in beekeeping should include studies on the potential side effects on CHC profiles, which may disrupt the

normal communication among members of the hive. We are currently investigating whether these laboratory findings are significant in the field when EO are applied in the beehives. To our knowledge, this is the first report of an effect of the ingestion of an essential oil on cuticular hydrocarbons of honeybees.

## Supporting information

**S1 Fig. Typical GCMS profile (total ion chromatogram, TIC) of CHC analyzed in this work.** Numbers of peaks are as in S1 and S2 Tables.
(TIF)

**S1 Table. Amount of CHC in honeybees fed on diets enriched with EOEb (2 EOEb were supplied: EOEb-2009 and EOEb-2014) in 3 concentrations (300, 3000 and 6000 ppm; Experiment I).** Results are shown as mean ± se (N = 4 per treatment).
(DOCX)

**S2 Table. Amount of CHC in honeybees with different health status (healthy and *Nosema*-infected; Experiment II), fed on different diets (EOEB-supplemented and control diets.** 2 EOEb were supplied: EOEb-2009 and EOEb-2014). Results are shown as mean ± se (N = 5 per treatment).
(DOCX)

**S3 Table. Comparison between individual CHC from control and control (ethanol) honeybees (univariate analysis results, P-values of the individual t-tests, no significant differences were found).** Analyses were run in the Metaboanalyst platform [44]. NI: non identified CHC, UK: unknowns -sum of NI-.
(DOCX)

**S4 Table. Comparisons of the individual 39 CHC in healthy vs *Nosema*-infected bees (2-factor-ANOVA, significant differences in bold).**
(DOCX)

## Acknowledgments

The authors would like to thank Sebastián Díaz and Yamandú Mendoza from the Instituto Nacional de Investigación Agropecuaria (INIA, Uruguay) for doing the collection of plant material.

## Author Contributions

**Conceptualization:** Carmen Rossini, Andrés González, Martín Javier Eguaras, Martín P. Porrini.

**Data curation:** Carmen Rossini.

**Formal analysis:** Carmen Rossini.

**Funding acquisition:** Carmen Rossini, Andrés González, Martín Javier Eguaras, Martín P. Porrini.

**Investigation:** Carmen Rossini, Federico Rodrigo, Belén Davyt, María Laura Umpiérrez, Antonella Cuniolo, Leonardo P. Porrini, Martín P. Porrini.

**Methodology:** Carmen Rossini, Federico Rodrigo, Belén Davyt, María Laura Umpiérrez, Paula Melisa Garrido, Martín P. Porrini.

**Project administration:** Carmen Rossini.

**Supervision:** Carmen Rossini, Martín P. Porrini.

**Validation:** Federico Rodrigo.

**Writing – original draft:** Carmen Rossini.

**Writing – review & editing:** Carmen Rossini, Martín P. Porrini.

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
