## [Decision Letter · Decision Letter 0]

10 Jul 2020

PONE-D-20-10247

Sub-lethal effects of the consumption of Eupatorium buniifolium essential oil in honeybees

PLOS ONE

Dear Dr. Rossini,

Thank you for submitting your manuscript to PLOS ONE. After careful consideration, we feel that it has merit but does not fully meet PLOS ONE’s publication criteria as it currently stands. Therefore, we invite you to submit a revised version of the manuscript that addresses the points raised by the two reviewers during the review process.

We look forward to receiving your revised manuscript.

Kind regards,

Nicolas Desneux

Academic Editor

PLOS ONE

Journal Requirements:

Reviewers' comments:

Reviewer's Responses to Questions

**Comments to the Author**

1. Is the manuscript technically sound, and do the data support the conclusions?

Reviewer #1: Yes

Reviewer #2: Yes

2. Has the statistical analysis been performed appropriately and rigorously? 

Reviewer #1: Yes

Reviewer #2: Yes

3. Have the authors made all data underlying the findings in their manuscript fully available?

Reviewer #1: Yes

Reviewer #2: Yes

4. Is the manuscript presented in an intelligible fashion and written in standard English?

Reviewer #1: Yes

Reviewer #2: No

5. Review Comments to the Author

Reviewer #1: General comments:

Rossini et al. assessed the potential side-effects of EO on nurse bees, which is a novel and important study. The experiments have been carried out well and the results are interesting, and the authors are testing something of significance which would be of interest to researchers who work on bee health and beekeepers. Going through the MS, I have one major concern on experimental setup: In experiment 1 the authors tested two types of EO in three final concentrations on healthy bees; however, only a single concentration 3000 ppm was tested for infected bees in experiment 2, and why not include the other two concentrations to get a complete view ? The results from infected bees are thought to be more relevant in practice.

Other minor comments:

Abstract is quite lengthy. I would suggest authors revising this part to be more concise but informative.

L 55-58: the sentence needs to be revised

L 63: it is unclear why EO in diet should be tested. Since EO is obviously volatile, the direct contact effects of EO on the bees could be more relevant. Clarify it.

L 48-78: the introduction section seems insufficiently introduced and poorly structured. The hypothesis that EO may have a sublethal effects on nurse bees could not be verified, and it is also unclear which biological aspects might be affected by EO treatment in diet, or other exposure risk of EO. I suggest authors enriching this section to be more informative ans specific in terms of content.

In addition, the topic of sublethal effects of pesticides on bees (as a broad) is not well introduced in this section. The authors should rely and cite the major review by Desneux et al. (2007) to introduce this scientific field in a clear way (e.g. sublethal effects vs. sublethal concentrations-doses, etc.). Papers on behavioral effects of pesticides on bees could be introduced here, e.g. Decourtye et al. (2004) and Han et al. (2010), as well as more general effects of pesticides on bees, such as Taning et al. (2019), and Varikou et al. (2019).

* Decourtye A, Devillers J, Cluzeau S, Charreton M, Pham-Delègue MH (2004) Effects of imidacloprid and deltamethrin on associative learning in honeybee under semi-field and laboratory conditions. Ecotoxicol Environ Saf 57:410–419

* Desneux N, Decourtye A, Delpuech JM (2007) The sublethal effects of pesticides on beneficial arthropods. Annu Rev Entomol 52:81–106

* Han P, Niu CY, Lei CL, Cui JJ, Desneux N (2010) Use of an innovative T-tube maze assay and the Proboscis Extension Response assay to assess sublethal effects of GM products and pesticides on learning capacity of the honey bee Apis mellifera L. Ecotoxicology 19:1612–1619

* Taning CNT, Vanommeslaeghe A, Smagghe G (2019) With or without foraging for food, field-realistic concentrations of sulfoxaflor are equally toxic to bumblebees (Bombus terrestris). Entomol Gen 39:151-155.

* Varikou K, Garantonakis N, Birouraki A (2019) Exposure of Bombus terrestris L. to three different active ingredients and two application methods for olive pest control. Entomol Gen 39:53-60

L 133: replace “from now on” with “hereafter referred to”

L 430: the references list should be update on the topic (using references . 38 is not directly linked to olfactory memory and learning performance, thus I suggested deleting it and using other references provided earlier for the introduction section). Moreover, the key review by Desneux et al. could be mentioned here as well.

L 467-469: why brackets were used?

Reviewer #2: The authors investigated the sub-lethal effects of the consumption of Eupatorium buniifolium essential oil in honeybees. I see there were a lot of experiments and the experiments were well controlled. However, I think the main problem of the paper is that the whole paper was not well organized. This is especially true for the result section. It is now hard to grasp the key point in the main text. I would suggest the authors reorganize the results to make it simpler and easier to understand, before receiving the review in more details.

Line 142, why the concentration of 3000ppm was used in experiment 2? Effects were found only at the dose of 6000ppm on CHC amounts of healthy bees in experiment 1. I would expect the dose of 6000ppm should be used in experiment2, instead of 3000ppm.

Line 217, I think the results is too long and not well organized. Furthermore the tables showing the CHC amount can be listed as supplementary tables, with only the compounds affected by the factors shown in the tables in maintext.

Line 252, 273 and other places, the names of the experiments were shown several times, which is very confusing on the writing structures.

I am not a native English speaker, but I feel the English of the paper needs more attention.

6. PLOS authors have the option to publish the peer review history of their article (what does this mean?). If published, this will include your full peer review and any attached files.

Reviewer #1: No

Reviewer #2: No

---

## [Author Response · Author response to Decision Letter 0]

11 Aug 2020

All answers below were also included in the cover letter.

We would like to sincerely thank all comments and suggestions. We had included most of them in our new manuscript, an as a result our work has considerably improved.

1) Reviewer #1:

a) Going through the MS, I have one major concern on experimental setup: In experiment 1 the authors tested two types of EO in three final concentrations on healthy bees; however, only a single concentration 3000 ppm was tested for infected bees in experiment 2, and why not include the other two concentrations to get a complete view ? The results from infected bees are thought to be more relevant in practice.

Answer: Since one of our purposes is to develop a varroacide whose active ingredient is the EOEb, the concentration to be used should be limited to that with no side effects. In order not to affect CHC, oral administration up to 3000 ppm appears to be safe (according with the results of experiment I). For this reason, we chose this concentration to be tested on Nosema-infected honeybees (in Experiment II). In this trial, there was also no effect of the EOEb consumption on the CHC of Nosema-infected bees. For that reason, we did not include all concentrations again. This point of view is now incorporated in the discussion section. 

b) Abstract is quite lengthy. I would suggest authors revising this part to be more concise but informative.

Answer: The abstract was shortened deleting the side results and methodological details.

c) L 55-58: the sentence needs to be revised

Answer: The sentence was rewritten.

d) L 63: it is unclear why EO in diet should be tested. Since EO is obviously volatile, the direct contact effects of EO on the bees could be more relevant. Clarify it.

Answer: The reviewer is right in that EOs are volatile, and that they are usually applied as fumigants. However, we have decided to test these products by ingestion to be sure that the putative sub-lethal effects could be detected, since toxicity of xenobiotics is usually greater through oral intake in honeybees (1). Besides, direct feeding of bees with health products can provide a more direct systemic way to exposure for the etiological agents to be treated as well as can circumvent issues related to variable fumigant volatility (2). 

This information was now added to the discussion section. 

e) L 48-78: the introduction section seems insufficiently introduced and poorly structured. The hypothesis that EO may have a sublethal effects on nurse bees could not be verified, and it is also unclear which biological aspects might be affected by EO treatment in diet, or other exposure risk of EO. I suggest authors enriching this section to be more informative ans specific in terms of content.

Answer: We have now changed our introduction, presenting the works we know about chronic effects of essential oils or thy components in hives (3-5), as well as some of the references suggested.

f) In addition, the topic of sublethal effects of pesticides on bees (as a broad) is not well introduced in this section. The authors should rely and cite the major review by Desneux et al. (2007) to introduce this scientific field in a clear way (e.g. sublethal effects vs. sublethal concentrations-doses, etc.). Papers on behavioral effects of pesticides on bees could be introduced here, e.g. Decourtye et al. (2004) and Han et al. (2010), as well as more general effects of pesticides on bees, such as Taning et al. (2019), and Varikou et al. (2019).

i) Decourtye A, Devillers J, Cluzeau S, Charreton M, Pham-Delègue MH (2004) Effects of imidacloprid and deltamethrin on associative learning in honeybee under semi-field and laboratory conditions. Ecotoxicol Environ Saf 57:410–419

ii) Desneux N, Decourtye A, Delpuech JM (2007) The sublethal effects of pesticides on beneficial arthropods. Annu Rev Entomol 52:81–106

iii) Han P, Niu CY, Lei CL, Cui JJ, Desneux N (2010) Use of an innovative T-tube maze assay and the Proboscis Extension Response assay to assess sublethal effects of GM products and pesticides on learning capacity of the honey bee Apis mellifera L. Ecotoxicology 19:1612–1619

iv) Taning CNT, Vanommeslaeghe A, Smagghe G (2019) With or without foraging for food, field-realistic concentrations of sulfoxaflor are equally toxic to bumblebees (Bombus terrestris). Entomol Gen 39:151-155.

v) Varikou K, Garantonakis N, Birouraki A (2019) Exposure of Bombus terrestris L. to three different active ingredients and two application methods for olive pest control. Entomol Gen 39:53-60

Answer: Some of these references were now properly included and discussed in the Introduction and the Discussion sections when corresponded. Besides, references from 2020 were also added (published after our first submission).

g) L 133: replace “from now on” with “hereafter referred to”

Answer: � The change was made in L133 as well as in all lines where “from now on” was used.

h) L 430: the references list should be update on the topic (using references . 38 is not directly linked to olfactory memory and learning performance, thus I suggested deleting it and using other references provided earlier for the introduction section). Moreover, the key review by Desneux et al. could be mentioned here as well.

Answer: � We do agree with the reviewer on that ref 38 is not linked to olfactory memory. However, we have used that reference to state that “Sub-lethal effects on adult honeybees include changes in their appetitive behavior…” which is the topic of ref. 38 in our first version (6) (Herbert LT, Vázquez DE, Arenas A, Farina WM. Effects of field-realistic doses of glyphosate on honeybee appetitive behaviour. J Exp Biol. 2014;217(19):3457-64). Therefore, we kept ref 38. However, we did include other references related to olfactory memory, as suggested by the reviewer, in this section as well as in the introduction. 

i) L 467-469: why brackets were used?

Answer: � Brackets were deleted.

2) Reviewer #2: ….. I think the main problem of the paper is that the whole paper was not well organized. This is especially true for the result section. It is now hard to grasp the key point in the main text. I would suggest the authors reorganize the results to make it simpler and easier to understand, before receiving the review in more details. 

Answer: � The result section was reorganized following a different scheme. It is now organized by the variables assessed instead by the experiments, in the following main sections:

1) Essential oil composition

2) Nosema ceranae development

3) Survival and food consumption

(a) Ethanol effect on survival and food consumption

(b) Effect of EOEb intake on survival and food consumption

4) Effect of the diets on CHC

(a) Overall results and effect of ethanol consumption on CHC

(b) Effect of EOEb ingestion at different doses on CHC of healthy bees (Experiment I)

(c) Effect of EOEb consumption on CHC from Nosema-infected honeybees (Experiment II)

a) Line 142, why the concentration of 3000ppm was used in experiment 2? Effects were found only at the dose of 6000ppm on CHC amounts of healthy bees in experiment 1. I would expect the dose of 6000ppm should be used in experiment2, instead of 3000ppm.

Answer: � This question is related to 1.a) above. Please see our answer above.

b) Line 217, I think the results is too long and not well organized. Furthermore the tables showing the CHC amount can be listed as supplementary tables, with only the compounds affected by the factors shown in the tables in maintext.

Line 252, 273 and other places, the names of the experiments were shown several times, which is very confusing on the writing structures.

Answer: � Results were re-organized; they are now presented by the variables controlled. The mention of the experiments was avoided. As a result, headings were changed (See also above, after (2)).

 � Table 1 is now included in the supplementary material as requested. 

c) I am not a native English speaker, but I feel the English of the paper needs more attention.

Answer: � The manuscript was corrected by a third party.

references related to the above answers:

1. Santos ACC, Cristaldo PF, Araujo APA, Melo CR, Lima APS, Santana EDR, et al. Apis mellifera (Insecta: Hymenoptera) in the target of neonicotinoids: A one-way ticket? Bioinsecticides can be an alternative. Ecotoxicol Environ Saf. 2018;163:28-36.

2. Ebert TA, Kevan PG, Bishop BL, Kevan SD, Downer RA. Oral toxicity of essential oils and organic acids fed to honey bees (Apis mellifera). J Apic Res. 2007;46(4):220-4.

3. Alayrangues J, Hotier L, Massou I, Bertrand Y, Armengaud C. Prolonged effects of in-hive monoterpenoids on the honey bee Apis mellifera. Ecotoxicology. 2016;25(5):856-62.

4. Bonnafé E, Drouard F, Hotier L, Carayon JL, Marty P, Treilhou M, et al. Effect of a thymol application on olfactory memory and gene expression levels in the brain of the honeybee Apis mellifera. Environmental Science and Pollution Research. 2015;22(11):8022-30.

5. Glavan G, Novak S, Bozic J, Jemec Kokalj A. Comparison of sublethal effects of natural acaricides carvacrol and thymol on honeybees. Pestic Biochem Physiol. 2020;166:104567.

6. Herbert LT, Vázquez DE, Arenas A, Farina WM. Effects of field-realistic doses of glyphosate on honeybee appetitive behaviour. J Exp Biol. 2014;217(19):3457-64.

---

## [Decision Letter · Decision Letter 1]

2 Oct 2020

PONE-D-20-10247R1

Sub-lethal effects of the consumption of Eupatorium buniifolium essential oil in honeybees

PLOS ONE

Dear Dr. Rossini,

Thank you for submitting your manuscript to PLOS ONE. After careful consideration, we feel that it has merit but does not fully meet PLOS ONE’s publication criteria as it currently stands. Therefore, we invite you to submit a revised version of the manuscript that addresses the points raised during the review process.

We look forward to receiving your revised manuscript.

Kind regards,

Nicolas Desneux

Academic Editor

PLOS ONE

Reviewers' comments:

Reviewer's Responses to Questions

**Comments to the Author**

1. If the authors have adequately addressed your comments raised in a previous round of review and you feel that this manuscript is now acceptable for publication, you may indicate that here to bypass the “Comments to the Author” section, enter your conflict of interest statement in the “Confidential to Editor” section, and submit your "Accept" recommendation.

Reviewer #1: All comments have been addressed

Reviewer #2: All comments have been addressed

2. Is the manuscript technically sound, and do the data support the conclusions?

Reviewer #1: Yes

Reviewer #2: Yes

3. Has the statistical analysis been performed appropriately and rigorously? 

Reviewer #1: Yes

Reviewer #2: Yes

4. Have the authors made all data underlying the findings in their manuscript fully available?

Reviewer #1: Yes

Reviewer #2: Yes

5. Is the manuscript presented in an intelligible fashion and written in standard English?

Reviewer #1: Yes

Reviewer #2: Yes

6. Review Comments to the Author

Reviewer #1: My previous comments have been answered and properly addressed by the authors. Upon reading the revised version, I found L 30-38 (in abstract) reads circuitous and they must be re-organized. Also the abstract should be one merged paragraph instead of several separate ones. Moreover, authors are suggested to be careful with the format requited by the journal; for example, I have see several small paragraphs in the “Statistical analyses” part. After revisions, I think the new version will be suitable for publishing in the journal.

Reviewer #2: The current version has been greatly improved by restructuring the results, thus I am happy to recommend for publication.

7. PLOS authors have the option to publish the peer review history of their article (what does this mean?). If published, this will include your full peer review and any attached files.

Reviewer #1: No

Reviewer #2: No

---

## [Author Response · Author response to Decision Letter 1]

16 Oct 2020

Comments from Reviewer #1 and answers:

1) Upon reading the revised version, I found L 30-38 (in abstract) reads circuitous and they must be re-organized. Also the abstract should be one merged paragraph instead of several separate ones. 

Answer: 

Abstract was edited. L 30-38 are now changed. 

The abstract is now one paragraph

2) Moreover, authors are suggested to be careful with the format requited by the journal; for example, I have see several small paragraphs in the “Statistical analyses” part. After revisions, I think the new version will be suitable for publishing in the journal.

Answer: 

“Statistical analyses” part was re-written.

The whole MS was also checked to fix format flaws. 

Reviewer #2 did not make any further suggestion.

---

## [Editor Report · Decision Letter 2]

20 Oct 2020

Sub-lethal effects of the consumption of Eupatorium buniifolium essential oil in honeybees

PONE-D-20-10247R2

Dear Dr. Rossini,

We’re pleased to inform you that your manuscript has been judged scientifically suitable for publication and will be formally accepted for publication once it meets all outstanding technical requirements.

Kind regards,

Nicolas Desneux

Academic Editor

PLOS ONE
---

## [Editor Report · Acceptance letter]

22 Oct 2020

PONE-D-20-10247R2 

Sub-lethal effects of the consumption of *Eupatorium buniifolium* essential oil in honeybees 

Dear Dr. Rossini:

I'm pleased to inform you that your manuscript has been deemed suitable for publication in PLOS ONE. Congratulations! Your manuscript is now with our production department. 

Kind regards, 

on behalf of

Dr. Nicolas Desneux 

Academic Editor

PLOS ONE